# How Predictors Affect Search Strategies in Neural Architecture Search?

## Abstract

Predictor-based Neural Architecture Search is an important topic since it can efficiently reduce the computational cost of evaluating candidate architectures. Most existing predictor-based NAS algorithms aim to design different predictors to improve the prediction performance. Unfortunately, even a promising performance predictor may suffer from the accuracy decline due to long-term and continuous usage, thus leading to the degraded performance of the search strategy. That naturally gives rise to the following problems: how predictors affect search strategies and how to appropriately use the predictor? In this paper, we take reinforcement learning (RL) based search strategy to study theoretically and empirically the impact of predictors on search strategies. We first formulate a predictor-RL-based NAS algorithm as model-based RL and analyze it with a guarantee of monotonic improvement at each trail. Then, based on this analysis, we propose a simple procedure of predictor usage, named $mixed\ batch$, which contains ground-truth data and prediction data. The proposed procedure can efficiently reduce the impact of predictor errors on search strategies with maintaining performance growth. Our algorithm, Predictor-based Neural Architecture Search with Mixed batch (PNASM), outperforms traditional NAS algorithms and prior state-of-the-art predictor-based NAS algorithms on three NAS-Bench-201 tasks and one NAS-Bench-ASR task .

## 1 Introduction

Neural Architecture Search (NAS) aims to automatically find out effective architectures in a pre-defined search space for a given dataset (Baker et al., 2016; Zoph & Le, 2016), which has shown to generate architectures that achieve promising results in many domains (Zoph et al., 2018; Tan & Le, 2019; Howard et al., 2019; Chen et al., 2020). However, due to the high computational cost for evaluating the generated architecture performance, traditional NAS methods are prohibitively costly in real-world deployment.

Recently, many approaches have been proposed to reduce the evaluation cost, which can be categorized into training-free predictors (Pham et al., 2018; Mellor et al., 2021) and training-based predictors (Wei et al., 2022; Springenberg et al., 2016; Shi et al., 2020; White et al., 2021a; Lu et al., 2021; Wen et al., 2020; Tang et al., 2020; Luo et al., 2018). Training-based methods, which require training a performance predictor to predict the final validation accuracy based on the feature of architecture, have received much more atten-

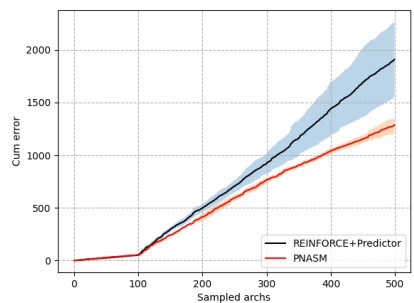

Figure 1: Cumulative error between true val and predicted val over sampled architectures. REINFORCE+Predictor means long and continuous usage of predictor without updating it.

tion due to their better generalization ability. Recent efforts on training-based methods focus on improving the prediction performance by designing models to precisely capture features of network architectures, e.g., GCN and Transformer. Several works demonstrate their robust predictions and combine them with the traditional search strategy such as Bayesian Optimization (BO) (Springenberg et al., 2016; Shi et al., 2020; White et al., 2021a) and Evolutionary Algorithms (EA) (Wei et al.,

2022; Lu et al., 2021; Wei et al., 2022). Unfortunately, even a promising performance predictor may suffer from the accuracy decline due to long-term and continuous usage (Fig. 1), thus leading to performance collapse. Most existing works barely consider the impact of predictor usage on the search strategy. The inappropriate usage of predictor may perform worse asymptotically than their predictor-free counterparts. That leads to two natural questions: how predictors affect search strategies and how to appropriately use the predictor to improve search efficiency?

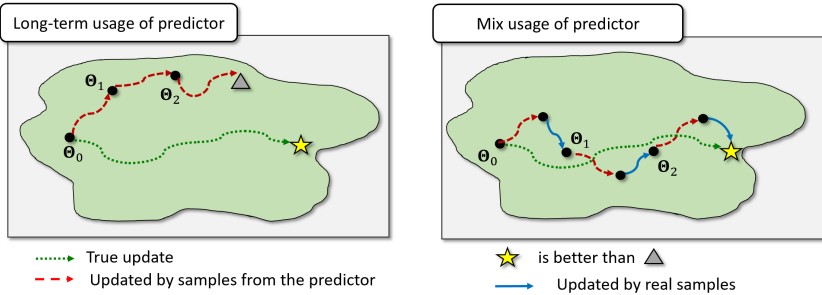

Figure 2: Parameters of policy updated by different ways. **Left**. Parameters of policy deviate far from the optimal one due to compounding error of long-term usage of predictor. **Right.** Limited mix usage of predictor can balance performance and computational cost.

In this paper, we take RL-based search strategy to study the impact of predictors on search strategies both theoretically and empirically. We first formulate a predictor-RL-based NAS algorithm as model-based RL and analyze a class of predictor-based NAS algorithms with improvement guarantees. Formula derivation results indicate that if the predictor is used for a long time, enlarged predictor error compounding policy error will lead to performance collapse. Then, based on the analysis, we propose a simple procedure of predictor usage, named $mixed\ batch$, to update the search strategy, which contains ground-truth data and prediction data. The prediction data, on the one side, can greatly improve sample efficiency, and on the other side encourages policy exploration. The ground-truth data allows the updated parameters close in parameter space and prevents a bad-update accidentally cause performance collapse (Fig. 2). We empirically demonstrate that the proposed procedure can achieve pronounced improvements in performance compared to other predictor-based NAS approaches.

To summarize, our contribution in this work are following:

- We conduct the first study of the impact of predictors on NAS search strategies both theoretically and empirically.

- We formulate and analyze a category of predictor-RL-based NAS algorithms with improvement guarantees based on predictor error and policy error. Theoretical analysis indicates that the long-term use of predictor declines the performance of search strategy.

- We propose a novel predictor-based NAS framework, namely PNASM (Predictor-based Neural Architecture Search with Mixed batch), to make limited usage of the performance predictor and improve the search performance.

- Our proposed method outperforms both traditional and predictor-based NAS methods and achieves state-of-the-art results on CIFAR-10, CIFAR-100, and ImageNet-16-120 of NAS-Bench-201, and TIMIT of NAS-Bench-ASR.

## 2 RELATED WORK

### 2.1 NEURAL ARCHITECTURE SEARCH

Traditional NAS methods, such as reinforcement learning (Zoph & Le, 2016; Baker et al., 2016), evolutionary search (Real et al., 2019), and gradient-based search (Liu et al., 2019), have shown to generate networks that outperform manually-designed networks. However, these algorithms require enormous search costs due to the high evaluation cost for generated architectures. To reduce

the search costs, researchers have proposed predictor-based methods to quickly estimate the performance of architectures instead of training from scratch (Wei et al., 2022; Springenberg et al., 2016; Shi et al., 2020; White et al., 2021a; Lu et al., 2021; Wen et al., 2020; Tang et al., 2020; Luo et al., 2018). There are two classes of predictor-based NAS methods:

**Training-based predictors.** Training-based predictors follow a supervised learning paradigm to learn the correlation between network architectures and their corresponding performance. These predictors are usually used within BO frameworks (Springenberg et al., 2016; Shi et al., 2020; White et al., 2021a), evolutionary frameworks (Wei et al., 2022; Lu et al., 2021; Wei et al., 2022), or without any search strategy (Wen et al., 2020), to conduct NAS. BONAS (Shi et al., 2020) adopts a GCN-based accuracy predictor as a surrogate function of BO to search for architectures. Similarly, BANANAS (White et al., 2021a) also uses BO to perform NAS and provides a thorough analysis of the relation between BO and the neural predictor. NPENAS (Wei et al., 2022) develops two kinds of neural predictors to guide evolutionary strategy to boost the exploration ability. NPNAS (Wen et al., 2020) directly uses a regression model to predict the validation accuracy of a large number of random architectures and chooses the top-k architectures to obtain the best one. SemiNAS (Tang et al., 2020) proposes a semi-supervised predictor to capture the intrinsic similarities of labeled and unlabeled architectures. TNASP (Lu et al., 2021) uses a Transformer-based predictor and evolutionary algorithms to perform NAS. Similar to these algorithms, our model PNASM also adopts the training-based predictor. Differently, we propose a novel update scheme by combining ground-truth data and prediction data to optimize the reinforcement learning strategy, which can reduce the impact of predictor error on the optimization strategy and improve search efficiency.

**Training-free predictors.** Recently, several works have proposed to compute statistics from a single minibatch data by a single forward and backward propagation. NASWOT (Mellor et al., 2021) evaluates randomly-initialized architectures based on binary activation codes of ReLU units. TE-NAS (Chen et al., 2021a) analyzes the spectrum of the neural tangent kernel and the number of linear regions in the input space to rank architectures. Zero-Cost NAS (Abdelfattah et al., 2021) compares six conventional reduced-training proxies to compute a model's score. Although these training-free predictors achieve satisfying results on some datasets, their performance cannot be guaranteed in practice due to the limited generalization ability (Lu et al., 2021; White et al., 2021b).

## 2.2 MODEL-BASED REINFORCEMENT LEARNING

Model-based Reinforcement Learning (MBRL) methods have shown great success on real-world sequential decision problems due to their sample efficiency ability (Kaelbling et al., 1996). MBRL learns a model of the environment, which predicts state transitions and rewards. Thus, they are widely used to solve problems where the data are hardly collected in real-world physical systems. The dynamics of environment are usually modeled by Gaussian processes (Deisenroth & Rasmussen, 2011), local linear models (Levine & Koltun, 2013; Kumar et al., 2016), and neural network function approximators (Draeger et al., 1995; Gal et al., 2016; Nagabandi et al., 2018; Janner et al., 2019; Yu et al., 2020; Shen et al., 2020). If we consider the evaluating of candidate architectures as an RL-environment, we can formulate predictor-RL-based NAS as MBRL. The difference between our formulation and the traditional MBRL is that our predictor is trained to predict rewards not the state transitions.

## 3 METHOD

### 3.1 PRELIMINARY

**NAS problem.** Given a dataset $\mathcal{D}$ and a search space $\mathcal{O}$ of neural architectures, the RL-based optimization strategy is to search the best architecture $A^* \in \mathcal{O}$ that maximizes the expected accuracy on the validation set $\mathcal{D}_{valid}$, which is defined by:

$$
\begin{aligned}
A^* = \underset{A \in \mathcal{O}}{\operatorname{argmax}} E_{(\mathcal{D}_{\text{train}}, \mathcal{D}_{\text{valid}}) \sim \mathcal{D}} \left[ \mathcal{R} \left( A_{\boldsymbol{w}^*}, \mathcal{D}_{\text{valid}} \right) \right] \\
\text{s.t. } \boldsymbol{w}^* = \underset{\boldsymbol{w}}{\operatorname{argmin}} \mathcal{L} \left( A_{\boldsymbol{w}}, \mathcal{D}_{\text{train}} \right)
\end{aligned}
\tag{1}
$$

where $\mathcal{R}(A_{\boldsymbol{w}^*}, \mathcal{D}_{valid})$ measures the accuracy of an architecture $A$ with parameters $\boldsymbol{w}^*$ on the validation data $\mathcal{D}_{valid}$. $\boldsymbol{w}$ is the parameters of architectures. $\mathcal{L}$ represents the loss of architectures on the training data $\mathcal{D}_{train}$.

**Predictor-based NAS.** Since evaluating an architecture in Eq. 1 typically takes hours, many NAS methods use performance predictors to speed up this process. A performance predictor $f_\phi$ generally consists of an encoder $f_E$ and a regressor $f_R$, which encodes the information of discrete architectures into continuous feature representations and learns the correlation from the network features and the network performance, respectively. Most training-based predictors are trained by the supervised learning from a database containing neural architectures $A$ and their corresponding performances $\mathcal{R}(A_{\boldsymbol{w}^*}, \mathcal{D}_{valid})$ (Luo et al., 2018; Wen et al., 2020; Chen et al., 2021b; Lu et al., 2021). That is, predictor $f_\phi$ is trained to minimize the MSE loss between the predicted accuracy and the true accuracy of the architecture sampled from the database:

$$\phi^* = \arg\min_\phi \sum_A \left( \mathcal{R}(A_{\boldsymbol{w}^*}, \mathcal{D}_{valid}) - f_\phi(A) \right)^2 \tag{2}$$

where $f_\phi(A)$ denotes the predicted performance of $A$. After the above process, the performance predictor can quickly predict the final accuracy or ranking of unseen architectures.

### 3.2 MBRL-BASED OPTIMIZATION STRATEGY

Predictor-RL-based NAS can be formulated as a MBRL problem with a tuple $\langle S, A, T, R, \gamma \rangle$, where $S, A, T, R$ and $\gamma$ denote the state space, the action space, the state transition dynamics, the reward function and the discount factor (Please see Section A of the Suppl. for more details). Normally, MBRL corresponds to recovering the state transition dynamics $T$ and the reward function $R$. Following the RL-based NAS framework (Zoph & Le, 2016; Baker et al., 2016; Zoph et al., 2018), we only need to recover the reward function $R$, which corresponds to the predictor $f_\phi$.

In this framework, an agent, also called the controller, samples $T$-step trajectory $\tau = (s_1, a_1, \ldots, s_T, a_T)$ at each episode, which corresponds to the description of a neural architecture $A = a_{1:T}$. Then, evaluate the performance of the generated architecture $A$ either by training from scratch or by the predictor. The evaluation result $R(\tau)$ is used as a reward signal to update the parameters $\theta$ of the policy $\pi$. After several iterations, the agent will learn to generate, with high probability, an architecture with high reward (accuracy). The goal of the agent is to maximize the expected reward:

$$J(\pi_\theta) = E_{\tau \sim \pi_\theta}[R(\tau)] = \sum_\tau R(\tau) p(\tau|\theta) \tag{3}$$

where $R(\tau)$ denotes the evaluated performance of the generated architecture $\tau$. $p(\tau|\theta)$ denotes the probability of a trajectory $\tau$.

As the reward signal $R$ is non-differentiable, one common approach is to use REINFORCE (Williams, 1992) to update the parameters $\theta$ of the policy:

$$\theta_{k+1} = \theta_k + \alpha \nabla_\theta J(\pi_\theta)|_{\theta_k} \tag{4}$$

where $\nabla_\theta J(\pi_\theta)$ is given by:

$$\nabla_\theta J(\pi_\theta) = E_{\tau \sim \pi_\theta} \left[ R(\tau) \nabla_\theta \log p(\tau|\theta) \right] \approx \frac{1}{N} \sum_{n=1}^{N} R(\tau^n) \nabla_\theta \log p(\tau^n|\theta)$$

$$= \frac{1}{N} \sum_{n=1}^{N} \sum_{t=1}^{T} \nabla_\theta \log \pi_\theta(a_t^n|s_t^n) R(\tau^n) \tag{5}$$

where $N$ is the number of neural architectures that the agent generates at each iteration (equivalent to the batch size) . $T$ is the number of candidate operations (actions) of a neural architecture.

### 3.3 MONOTONIC PREDICTOR-BASED IMPROVEMENT

In this section, we will give a monotonic improvement based on a general predictor-RL-based NAS as described in Algorithm 1, where the policy is optimized based on the data provided by the predictor. The performance of policy is affected by the predictor usage since errors in the predictor may be exploited by the policy optimization, thus leading to a large gap between the true performance of the policy and that under the predictor.

---

**Algorithm 1** Monotonic Predictor-Based Policy Optimization.

---

**Input:** # of initial samples $S$; Batch-size $N$
 1: Collect $S$ architectures by running policy $\pi_\theta$
 2: Evaluate $S$ architectures by training from scratch and store the samples into $\mathcal{D}_t$
 3: Initialize predictor $f_\phi$ and policy $\pi_\theta$ by the collected samples
 4: **while** time limit not exceeded **do**
 5:     Collect $N$ architectures by running policy $\pi_\theta$
 6:     Evaluate $N$ architectures by the predictor $f_\phi$ and store the samples into $\mathcal{B}$
 7:     Update the policy $\pi_\theta$ via Eq. 4 with $\mathcal{B}$;
 8: **end while**

---

Our goal is to build a performance guarantee for the predictor-RL-based NAS. Motivated by MBPO (Janner et al., 2019), we wish to construct a lower bound of the following form:

$$\eta(\pi) \geq \eta(\hat{\pi}) - C \tag{6}$$

where $\eta(\pi)$ represents the expected true performance of policy $\pi$ which are updated by the reward signal of training architectures from scratch, i.e., in the true dynamics; whereas $\eta(\hat{\pi})$ denotes the expected performance of policy $\pi$ that are updated based on the reward signal provided by the predictor. Such a statement guarantees that, as long as we improve by at least $C$ under the performance predictor, we can guarantee improvement over the true performance $\eta$.

The difference $C$ between the true performance and that under the predictor comes from two error quantities of the performance predictor: generalization error due to the prediction ability, and policy error (distribution shift) due to the updated policy receiving the reward signal provided by the predictor. Since the performance predictor is trained using supervised learning, we define this generalization error $\epsilon_m$ by:

$$\max_{\tau \sim \pi_D} |R_1(\tau) - R_2(\tau)| \leq \epsilon_m \tag{7}$$

where $R_1(\tau)$ denotes the true performance of an architecture described by $\tau$ and $R_2(\tau)$ denotes the prediction performance under the performance predictor, i.e., $R_2 := f_\phi$; $\pi_D$ denotes the data-collecting policy. This error $\epsilon_m$ can be estimated in practice by measuring the difference between the true reward and the prediction reward on the same trajectory $\tau$, which is obtained under the data-collecting policy $\pi_D$. We define policy error by the maximum total-variation distance of the policy between iterations:

$$\max_s D_{TV}(\pi(a|s)||\pi_D(a|s)) \leq \epsilon_\pi \tag{8}$$

In practice, we can measure the KL divergence between policies. Based on these two sources of errors (generalization error $\epsilon_m$ and policy error $\epsilon_\pi$), we now give our bound:

**Theorem 1** *Let the generation error between the true reward and the prediction reward be bounded at each trajectory by $\epsilon_m$ and the policy divergence be bounded by $\epsilon_\pi$. Then the expected true reward and expected prediction reward of the policy are bounded as:*

$$\eta(\pi) \geq \eta(\hat{\pi}) - \underbrace{(\sum_{\tau=1}^{N} 2R_{max}\epsilon_\pi + \sum_{\tau=1}^{N} \epsilon_m p(\tau|\widetilde{\theta}))}_{C(\epsilon_m, \epsilon_\pi)} \tag{9}$$

*Proof.* See Appendix Theorem B.1.

This bound implies that as long as we improve the expected reward $\eta(\hat{\pi})$ under the predictor by more than $C(\epsilon_m, \epsilon_\pi)$, we can guarantee improvement under the expected true reward.

### 3.4    MIXING REAL-BASED AND PREDICTOR-BASED UPDATES

Theorem 1 provides a useful relationship between true rewards and prediction rewards. However, it is noted that if the predictor error $\epsilon_m$ is too high, there may not exist a policy that can guarantee the improvement. Besides, the analysis of Theorem 1 relies on using the prediction reward to update the policy continuously, i.e., equivalent to increasing $N$, which allows model error to compound with

policy error and results in a large gap value $C$. Thus, we can improve the algorithm by relying less on the performance predictor when the performance predictor is inaccurate and instead by training neural architectures to rely more on real data.

For the above issues, we introduce a simple procedure *mixed batch* to reduce the influence of two errors on the policy. A policy with *mixed batch*, denoted as $\pi_{mix}$, means a batch of $N$ samples are collected by the following two steps: first, run policy $\pi$ to generate first $k$ architectures which are evaluated by training from scratch (under the true environment); then, generate $N - k$ architectures under the learned performance predictor $f_\phi$ (Algorithm 2). Under this scheme, the expected reward can be bounded as follows:

**Theorem 2** *Given the expected reward $\eta(\pi_{mix})$ from the k-steps mixed batch method, we have*

$$\eta(\pi) \geq \eta(\pi_{mix}) - \left[ \sum_{\tau=1}^{N} R_{max}\epsilon_\pi + \sum_{\tau=k+1}^{N} R_{max}\epsilon_\pi + \sum_{\tau=k+1}^{N} \epsilon_m p(\tau|\widetilde{\theta}) \right] \quad (10)$$

*Proof.* See Appendix Theorem B.2.

This bound implies that as long as we mix the true data and prediction data into one batch, we can reduce the error caused by long-term use of the performance predictor.

---

**Algorithm 2** Predictor-based Neural Architecture Search with Mixed batch.

---

**Input:** # of initial samples $S$; batch size $N$; # of true samples $k$;
 1: Collect $S$ architectures by running policy $\pi_\theta$
 2: Evaluate $S$ architectures by training from scratch and store the samples into $\mathcal{D}_t$
 3: Initialize predictor $f_\phi$ and policy $\pi_\theta$ by the collected $S$ samples
 4: **while** time limit not exceeded **do**
 5:     Collect $k$ architectures by running policy $\pi_\theta$
 6:     Evaluate $k$ architectures by training from scratch and store the samples into $\mathcal{D}_t$
 7:     Collect $N - k$ architectures by running policy $\pi_\theta$
 8:     Evaluate $N - k$ architectures by the predictor $f_\phi$ and store the samples into $\mathcal{D}_p$;
 9:     Select $k$ and $N - k$ pairs from $\mathcal{D}_t$ and $\mathcal{D}_p$ respectively to form a mini-batch $\mathcal{B}$;
10:     Update the policy $\pi_\theta$ via Eq. 4 with $\mathcal{B}$;
11:     Retrain the predictor $f_\phi$ via Eq. 2 with $\mathcal{D}_t$;
12: **end while**

---

## 4 EXPERIMENTS

We employ our model on four datasets, specifically NAS-Bench-201: CIFAR-10, CIFAR-100, and ImageNet-120, NAS-Bench-ASR: TIMIT. We split up our experiments into three categories: selecting the best predictor for search spaces, evaluating the performance of our model and other popular algorithms on two NAS benchmarks, and performing ablation experiments. Moreover, we put several experimental results; the implementation details; information of baselines in Appendix D .

### 4.1 CHOOSE PREDICTOR FOR NAS

To obtain high-performance predictor-RL-based NAS algorithm, we first choose a high-performance predictor among the currently most popular performance predictors, including MLP, GCN, BA-NANAS, BONAS, NAO, SemiNAS, and Transformer. We randomly run 20 times for each predictor on CIFAR-10 and report the mean and variance of the Kendall's Tau correlation coefficient on test samples. Kendall's Tau is a common indicator measuring the correlation between the ranking of prediction values and the true labels, and higher value indicates more accurate prediction. The training and testing samples are sampled randomly from the collected architecture-accuracy pairs. Table 1 presents the comparison results, from which we can make the following observations: 1) As the number of training samples increases, the performance of predictor improves. Therefore, it's important for all predictors to have enough initial training samples. 2) SemiNAS and Transformer perform well even with a small number of training samples. For example, their Kendall's Tau can achieve around 0.550 when the number of training samples is 100 or 200. According to the experimental results, we choose SemiNAS as our performance predictor on NAS-Bench-201 since it

adopts semi-supervised learning (Tang et al., 2020) to train the predictor, which could make full use of the unlabeled architecture information as the size of training samples increases, thus allowing it to outperform Transformer.

Table 1: Performance Comparisons of Predictors on CIFAR-10.

| Training Samples | 100 | 200 | 400 | 600 | 800 | 1000 |
|---|---|---|---|---|---|---|
| Test Samples | 200 | 200 | 200 | 200 | 200 | 200 |
| MLP | $0.350 \pm 0.06$ | $0.417 \pm 0.05$ | $0.500 \pm 0.05$ | $0.573 \pm 0.03$ | $0.636 \pm 0.04$ | $0.671 \pm 0.03$ |
| GCN | $0.509 \pm 0.09$ | $0.570 \pm 0.04$ | $0.609 \pm 0.07$ | $0.604 \pm 0.06$ | $0.624 \pm 0.07$ | $0.613 \pm 0.09$ |
| BANANAS | $0.318 \pm 0.06$ | $0.404 \pm 0.06$ | $0.506 \pm 0.03$ | $0.577 \pm 0.03$ | $0.614 \pm 0.02$ | $0.664 \pm 0.02$ |
| BOHAMIANN | $0.435 \pm 0.08$ | $0.486 \pm 0.06$ | $0.525 \pm 0.04$ | $0.529 \pm 0.04$ | $0.529 \pm 0.04$ | $0.535 \pm 0.04$ |
| BONAS | $0.329 \pm 0.10$ | $0.422 \pm 0.08$ | $0.480 \pm 0.08$ | $0.466 \pm 0.06$ | $0.491 \pm 0.07$ | $0.519 \pm 0.04$ |
| NAO | $0.486 \pm 0.05$ | $0.512 \pm 0.06$ | $0.536 \pm 0.05$ | $0.556 \pm 0.02$ | $0.575 \pm 0.03$ | $0.587 \pm 0.03$ |
| SemiNAS | $0.546 \pm 0.05$ | $0.588 \pm 0.02$ | $0.635 \pm 0.02$ | $\mathbf{0.669 \pm 0.02}$ | $\mathbf{0.687 \pm 0.02}$ | $\mathbf{0.704 \pm 0.02}$ |
| Transformer | $\mathbf{0.574 \pm 0.05}$ | $\mathbf{0.620 \pm 0.03}$ | $\mathbf{0.646 \pm 0.02}$ | $0.664 \pm 0.02$ | $0.680 \pm 0.02$ | $0.686 \pm 0.03$ |

## 4.2 EMPIRICAL RESULTS

**Performance on NAS-Bench-201.** Table 2 shows the comparison between the proposed method and strong baselines. According to the time budget, we randomly run 100 and 20 times for traditional and predictor-based methods, respectively. "search" means the total search time (time budget) including the time for initializing the predictor. "+B" and "+E" denote that we combine the predictors with optimization strategies of Bayesian Optimization and Evolutionary algorithm, respectively. The best result on each dataset is in boldface. We can easily see from the experimental results that: 1) Our method (PNASM) outperforms both traditional and predictor-based methods and achieves the performance of 94.33%, 72.89%, and 46.44% on the three datasets, respectively, close to the optimal performance. 2) Compared with REINFORCE, our method improves the test accuracy on CIFAR-10, CIFAR-100, and ImageNet-16-120 by 0.31%, 0.54%, and 0.7%, respectively, which demonstrates that the appropriate usage of predictor can help the optimization strategy to explore the more promising space. 3) Compared with the advanced SemiNAS+E, our PNASM (SemiNAS+RL) improves the validation accuracy on CIFAR-10, CIFAR-100, and ImageNet-16-120 by 0.06%, 0.24%, and 0.39%, respectively. The experiment results suggest that compared with EA, the advantage achieved by RL-based optimization strategy lies in the proper usage of predictor, i.e., *mixed batch* which contains true architecture-accuracy pairs and predicted ones, thus leading to the improvement in performance. 4) The performance of all predictors combined with BO is worse than those combined with EA and BOHB, which is consistent with the experiment result in White et al. (2021b).

Table 2: Performance Comparisons of NAS methods on CIFAR-10, CIFAR-100, and ImageNet-16-120. "Optimal value" indicates the highest accuracy achieved on NAS-Bench-201.

| Method | CIFAR-10 | | | CIFAR-100 | | | ImageNet-16-120 | | |
|---|---|---|---|---|---|---|---|---|---|
| | search(s) | validation | test | search(s) | validation | test | search(s) | validation | test |
| | | | | Traditional Method | | | | | |
| REA | 40000 | $91.39 \pm 0.22$ | $94.17 \pm 0.24$ | 75000 | $72.73 \pm 0.73$ | $72.50 \pm 0.63$ | 200000 | $46.03 \pm 0.57$ | $46.01 \pm 0.65$ |
| RS | 40000 | $91.17 \pm 0.25$ | $93.96 \pm 0.25$ | 75000 | $71.81 \pm 0.87$ | $71.89 \pm 0.85$ | 200000 | $45.18 \pm 0.81$ | $45.41 \pm 0.91$ |
| REINFORCE | 40000 | $91.27 \pm 0.21$ | $94.02 \pm 0.22$ | 75000 | $72.38 \pm 0.63$ | $72.35 \pm 0.55$ | 200000 | $45.65 \pm 0.58$ | $45.74 \pm 0.77$ |
| BOHB | 51000 | $91.34 \pm 0.21$ | $94.10 \pm 0.23$ | 98000 | $72.62 \pm 0.81$ | $72.42 \pm 0.65$ | 290000 | $45.98 \pm 0.50$ | $46.11 \pm 0.63$ |
| | | | | Predictor-based Method | | | | | |
| MLP+B | 36000 | $91.22 \pm 0.23$ | $94.04 \pm 0.22$ | 71000 | $72.16 \pm 0.88$ | $72.32 \pm 0.91$ | 210000 | $45.47 \pm 0.61$ | $45.63 \pm 0.79$ |
| MLP+E | 39000 | $91.37 \pm 0.25$ | $94.19 \pm 0.20$ | 75000 | $72.67 \pm 0.34$ | $72.58 \pm 0.35$ | 230000 | $46.08 \pm 0.36$ | $45.87 \pm 0.60$ |
| GCN+B | 39000 | $91.15 \pm 0.20$ | $93.92 \pm 0.17$ | 74000 | $72.22 \pm 0.79$ | $72.21 \pm 0.84$ | 210000 | $45.19 \pm 0.56$ | $45.76 \pm 0.77$ |
| GCN+E | 40000 | $91.45 \pm 0.12$ | $94.20 \pm 0.23$ | 79000 | $72.94 \pm 0.50$ | $72.75 \pm 0.48$ | 230000 | $46.12 \pm 0.29$ | $45.93 \pm 0.55$ |
| BANANAS+B | 36000 | $91.15 \pm 0.20$ | $93.91 \pm 0.23$ | 70000 | $71.93 \pm 0.89$ | $71.96 \pm 0.94$ | 200000 | $45.47 \pm 0.60$ | $45.73 \pm 0.61$ |
| BONAS+B | 36000 | $91.16 \pm 0.23$ | $93.96 \pm 0.23$ | 70000 | $71.69 \pm 0.81$ | $71.67 \pm 0.81$ | 210000 | $45.42 \pm 0.62$ | $46.02 \pm 0.68$ |
| NAO+B | 37000 | $91.27 \pm 0.15$ | $94.08 \pm 0.17$ | 72000 | $72.14 \pm 0.62$ | $72.07 \pm 0.63$ | 210000 | $45.46 \pm 0.65$ | $45.88 \pm 0.71$ |
| NAO+E | 40000 | $91.40 \pm 0.26$ | $94.23 \pm 0.27$ | 78000 | $72.90 \pm 0.38$ | $72.62 \pm 0.40$ | 230000 | $45.78 \pm 0.60$ | $45.73 \pm 0.73$ |
| SemiNAS+B | 37000 | $91.17 \pm 0.22$ | $93.88 \pm 0.24$ | 72000 | $72.17 \pm 0.43$ | $72.10 \pm 0.66$ | 210000 | $45.13 \pm 0.74$ | $45.24 \pm 0.71$ |
| SemiNAS+E | 40000 | $91.45 \pm 0.19$ | $94.26 \pm 0.23$ | 76000 | $72.84 \pm 0.55$ | $72.77 \pm 0.53$ | 230000 | $45.95 \pm 0.38$ | $45.67 \pm 0.54$ |
| PNASM(Ours) | 40000 | $\mathbf{91.51 \pm 0.05}$ | $\mathbf{94.33 \pm 0.06}$ | 75000 | $\mathbf{73.08 \pm 0.22}$ | $\mathbf{72.89 \pm 0.36}$ | 200000 | $\mathbf{46.34 \pm 0.08}$ | $\mathbf{46.44 \pm 0.07}$ |
| PNASM-A(Ours) | 40000 | $91.48 \pm 0.10$ | $94.26 \pm 0.12$ | 65000 | $72.85 \pm 0.42$ | $72.68 \pm 0.49$ | 200000 | $46.09 \pm 0.30$ | $45.94 \pm 0.69$ |
| **Optimal value** | - | 91.61 | 94.37 | - | 73.49 | 73.51 | - | 46.73 | 47.31 |

**Performance on NAS-Bench-ASR.** Table 3 shows the comparison between the proposed method and traditional NAS algorithms. Since there is no information about the time cost of each architecture on NAS-Bench-ASR, we terminate the search process as the number of sampled true architectures reaches 300. We randomly run 20 times for each method. The best result is in boldface. We can easily see from the experimental results that: 1) Our model (PNASM) achieves the best results.

2) Compared with REINFORCE, our method improves the validation PER by 0.12%. 3) REA performs well compared to other traditional NAS algorithms. As we can see from the above results, our model still performs well on other NAS benchmarks, which indicates that *mixed batch* does help the RL-based search strategy make full use of the predictor.

Table 3: Performance Comparisons of NAS methods on NAS-Bench-ASR.

| Method | Sampled True Archs | Val. PER(%) | Test PER(%) |
|---|---|---|---|
| REA | 300 | $19.13 \pm 0.12$ | $21.76 \pm 0.36$ |
| RS | 300 | $19.17 \pm 0.08$ | $21.91 \pm 0.29$ |
| REINFORCE | 300 | $19.21 \pm 0.13$ | $21.82 \pm 0.29$ |
| PNASM(Ours) | 300 | $\mathbf{19.09 \pm 0.10}$ | $\mathbf{21.73 \pm 0.21}$ |
| PNASM-A(Ours) | 300 | $19.10 \pm 0.09$ | $21.74 \pm 0.17$ |

## 4.3 ABLATION STUDY

**Impact of Number of True Samples** $k$**.** The value of $k$ determines the ratio of true samples to prediction ones in a batch, which balances the performance and the computational costs. To study the impact of value $k$ on the final performance, we conduct a series of experiments on the three datasets with different values of $k$. Each experiment samples 1000 architectures. "search" means the total time costs including the time of predictor initialization .

Table 4 shows the results of PNASM with different $k$ values over 20 runs with different seeds, from which we can see that: 1) The model without using the predicted data ($k$=all) performs well on the three datasets, but incurs a large computational cost. Conversely, if we use the predictor all the time ($k$=0), the model performs poorly on the three datasets but with least computational cost, which demonstrate that a long term usage of predictor will amplify both the predictor error $\epsilon_m$ and the policy error $\epsilon_\pi$, thus leading to policy collapse. 2) A specific value of $k$ can achieve comparable performance to that $k$=all. For example, with the setting of $k$=5, the model achieves the performance 91.50% on the validation dataset on CIFAR-10. Similarly, $k$=2 on CIFAR-100 and $k$=2 on ImageNet-16-120, which demonstrates that $mixed\ batch$ is effective way of using predictor, which allows the search strategy to maintain excellent performance with less computational cost. Compared to the model with $k$=all, the model with $k$=5 achieves around $2\times$ speedups on CIFAR-10, and the model with $k$=2 brings around $4\times$ speedups on CIFAR-100 and ImageNet-16-120. 3) Models with $k$=5, 8, and 15, outperform that with $k$=15 on CIFAR-100. We speculate the reason of this phenomenon is that the usage of predictor injects noise into the parameter space of policy. The parameter noise limited in a reasonable range allows the policy to better explore the search space, as indicated by (Fortunato et al., 2018; Plappert et al., 2018).

To further study how $k$ affect the search strategy during the search process, Fig. 3 presents the current best architecture's validation accuracy, from which we can make the following observations: 1) The performance of $k = 0$ (long-term and continuous predictor usage) is inferior to others in most cases, which demonstrates that the long-term usage of unreliable predictor further exacerbates policy error. 2) As the number of ground-truth data increases ($k$ from 2 to 15), the performance curve gets close to that of $k$=all (blue line). In particular, there is a specific value $k$ can achieve comparable validation accuracy to that of $k$=all, e.g., $k$=5, with low computational cost. Therefore, $k$=5 is a good value that balances the performance and the computational costs. We recommend the true samples in the $mixed\ batch$ is 5 for other datasets.

**Adaptive Method.** Although $k$ can achieve a good trade-off between the performance and the time cost, but the proper setting of $k$ requires precisely fine-tuning. To simplify the setting, we propose an adaptive method (PNASM-A), which can dynamically adjust $k$ by measuring the differences between policies in two consecutive iterations:

$$k = \alpha \times N \tag{11}$$

where $\alpha$ is given by:

$$\alpha = \begin{cases} D_{KL}(\pi_{i-1}, \pi_i), & 0 \leq D_{KL}(\pi_{i-1}, \pi_i) < 1 \\ 1, & 1 \leq D_{KL}(\pi_{i-1}, \pi_i) \end{cases} \tag{12}$$

Table 4: Comparison of PNASM with different $k$ values on NAS-Bench-201.

| True samples($k$) | $k$=0 | $k$=2 | $k$=5 | $k$=8 | $k$=15 | $k$=all |
|---|---|---|---|---|---|---|
| | | | CIFAR-10 | | | |
| validation | $91.23 \pm 0.20$ | $91.37 \pm 0.22$ | $\mathbf{91.50 \pm 0.06}$ | $91.50 \pm 0.05$ | $91.40 \pm 0.14$ | $91.50 \pm 0.04$ |
| test | $93.98 \pm 0.22$ | $94.06 \pm 0.26$ | $94.28 \pm 0.16$ | $94.26 \pm 0.16$ | $94.19 \pm 0.19$ | $94.31 \pm 0.05$ |
| search(s) | 17000 | 31000 | 49000 | 69000 | 110000 | 130000 |
| | | | CIFAR-100 | | | |
| validation | $72.18 \pm 0.68$ | $72.99 \pm 0.36$ | $73.09 \pm 0.12$ | $\mathbf{73.14 \pm 0.16}$ | $73.02 \pm 0.13$ | $72.99 \pm 0.10$ |
| test | $72.14 \pm 0.52$ | $72.84 \pm 0.41$ | $72.79 \pm 0.42$ | $72.77 \pm 0.40$ | $72.44 \pm 0.32$ | $72.36 \pm 0.24$ |
| search(s) | 34000 | 59000 | 96000 | 130000 | 220000 | 260000 |
| | | | ImageNet-16-120 | | | |
| validation | $45.85 \pm 0.82$ | $\mathbf{46.30 \pm 0.23}$ | $46.21 \pm 0.29$ | $46.21 \pm 0.16$ | $46.32 \pm 0.00$ | $46.31 \pm 0.02$ |
| test | $45.84 \pm 0.92$ | $46.31 \pm 0.35$ | $46.41 \pm 0.36$ | $45.93 \pm 0.66$ | $46.47 \pm 0.00$ | $46.47 \pm 1.35$ |
| search(s) | 100000 | 170000 | 280000 | 390000 | 640000 | 780000 |

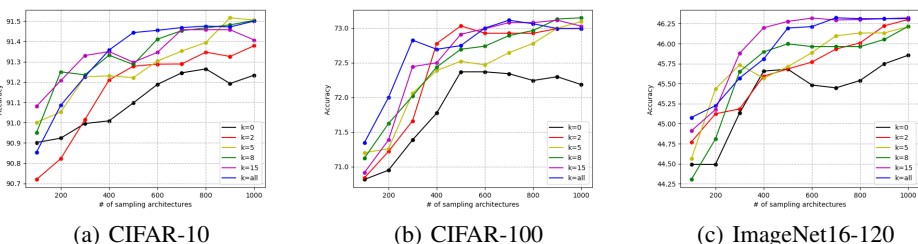

(a) CIFAR-10      (b) CIFAR-100      (c) ImageNet16-120

Figure 3: Validation accuracy vs. number of architectures on different settings of $k$.

where $\pi_{i-1}$ represents the policy after $i - 1^{th}$ iteration; $\pi_i$ denotes the policy after $i^{th}$ iteration. For the $i + 1^{th}$ iteration, $k = \alpha N$. As Table 2 shown, the adaptive method (PNASM-A) still outperforms other baselines on all three datasets. Besides, the performance of PNASM-A is close to that of PNASM, which demonstrate the effective of our adaptive variant.

## 5 CONCLUSION AND FUTURE WORK

In this paper, we investigate the role of predictor usage in neural architecture search procedures both theoretically and empirically. We first formulate predictor-RL-based NAS as model-based RL problem, and provide it with monotonic improvement guarantees, which suggests that the long-term and continuous usage of predictor will degrade the performance due to the model error exploited by the search policy. Motivated by this analysis, we then propose a novel framework PNASM that uses a special procedure, $mixed\ batch$, to justify predictor usage, which can mitigate the impact of predictor errors on search strategies and reduce the computational cost. Extensive experiments on NAS-Bench-201 have shown the effectiveness of the proposed method. In the future, we plan to investigate how to appropriately use the predictor with other search strategies.

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

# A WORKFLOW OF GENERATING ARCHITECTURE BY RL-AGENT

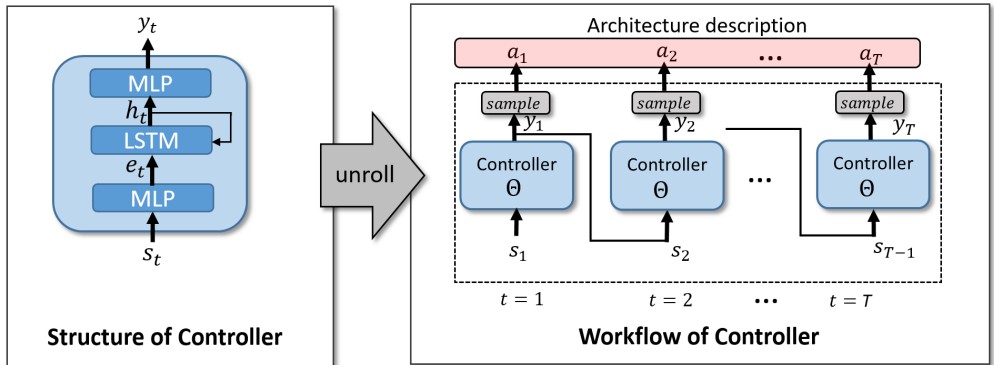

Figure 4: **Left**. Internal structure of the controller. **Right.** Workflow of the controller. It unrolls $T$ steps to output a trajectory $\tau = \{s_1, a_1, \ldots, s_T, a_T\}$, which describes an architecture.

Figure 4 presents the main structure of the controller. The controller consists of two multilayer perceptron (MLP) layers which serve as the input-embedding layer and the output-embedding layer, and an LSTM network which is the core of the controller for remembering previous decisions. At each episode, the controller unrolls $T$ time-steps to sample a trajectory $\tau = \{s_1, a_1, \ldots, s_T, a_T\}$. $\tau$ describes the representation of an architecture. At each step $t$, it works as follows:

The input $s_t$ is fed into the input-embedding layer, which converts $s_t$ into a high-dimensional embedding $e_t$, thus making the agent better observe the state:

$$e_t = W_{in} \cdot s_t + b_{in} \tag{13}$$

where $W_{in}$ and $b_{in}$ are embedding parameters of the input layer.

Then, $e_t$ is fed to the core network consisting of LSTM layer, which helps the agent explore the correlation between decisions:

$$o_t, h_t = LSTM(e_t, h_{t-1}) \tag{14}$$

Next, the output of LSTM $o_t$ is fed to the output layer to convert the output of the LSTM into a low-dimensional representation $y_t$, which denotes the distribution of the candidate operations, where $y_t = [\mu_t, \sigma_t]$.

$$y_t = W_{out} \cdot o_t + b_{out} \tag{15}$$

where $W_{out}$ and $b_{out}$ are embedding parameters of the output layer.

At last, given the distribution for candidate operations, the agent samples one by the same sampling technique:

$$a_t = Sample(\mathcal{N}(\mu_t, \sigma_t)) \tag{16}$$

$y_t$ is fed to the state at the the next time step $t + 1$:

$$s_{t+1} = y_t \tag{17}$$

The initial state $s_1$ is a zero embedding vector.

Under the special workflow of the controller, the probability of trajectory is equal to:

$$p(\tau|\theta) = p(s_1)\,\pi_\theta\,(a_1|s_1)\,p(s_2|s_1,a_1)\,\pi_\theta\,(a_2|s_2)\,p(s_3|s_2,a_2)\cdots\pi_\theta\,(a_T|s_T)$$
$$= p(s_1)\prod_{t=1}^{T}\pi_\theta\,(a_t|s_t)\,p(s_{t+1}|s_t,a_t) \tag{18}$$

Since the next state $s_{t+1}$ is equal to $y_t$, which is converted by the previous action $a_t$, both $p(s_1)$ and $p(s_{t+1}|s_t,a_t)$ are equal to one. Thus, $p(\tau|\theta)$ is simplified as:

$$p(\tau|\theta) = \prod_{t=1}^{T}\pi_\theta\,(a_t|s_t) \tag{19}$$

Therefore, the RL-based NAS can be formulated as MDP with fixed transition probability but unknown reward function.

## B THEOREMS

**Theorem B.1** *Monotonic predictor-based improvement:*

$$\eta(\pi) \geq \eta(\hat{\pi}) - \left[\sum_{\tau=1}^{N} 2R_{max}\epsilon_\pi + \sum_{\tau=1}^{N}\epsilon_m p(\tau|\widetilde{\theta})\right] \tag{20}$$

*Proof. Let $\pi_D$ denote the data collecting policy (old policy under the true environment). Since the performance predictor relies on the training data collected by the policy $\pi_D$, we need to introduce $\pi_D$ by adding and subtracting $\eta(\pi_D)$, to get:*

$$\eta(\pi) - \eta(\hat{\pi}) = \underbrace{\eta(\pi) - \eta(\pi_D)}_{L_1} + \underbrace{\eta(\pi_D) - \eta(\hat{\pi})}_{L_2}$$

*We can bound $L_1$ and $L_2$ both using Lemma C.2.*
*For $L_1$, there is no predictor error (generation error) since $\pi$ and $\pi_D$ run under the true environment:*

$$L_1 \geq -\sum_{\tau=1}^{N} R_{max}\epsilon_\pi$$

*For $L_2$, policy $\hat{\pi}$ runs under the predictor model which incurs predictor error and policy error. Thus, we have:*

$$L_2 \geq -\sum_{\tau=1}^{N} R_{max}\epsilon_\pi - \sum_{\tau=1}^{N}\epsilon_m p(\tau|\widetilde{\theta})$$

*The desired result is obtained by adding the two bounds together.*

**Theorem B.2** *Mixed batch bound:*

$$\eta(\pi) \geq \eta(\pi_{mix}) - \left[\sum_{\tau=1}^{N} R_{max}\epsilon_\pi + \sum_{\tau=k+1}^{N} R_{max}\epsilon_\pi + \sum_{\tau=k+1}^{N}\epsilon_m p(\tau|\widetilde{\theta})\right] \tag{21}$$

*Proof. Let $\pi_{mix} := \pi_D, \hat{\pi}$ denote the policy with the mixed batch which runs the old policy $\pi_D$ under the true dynamics until $k$ samples, then executes the new policy $\pi$ under the predictor in the last $N-k$ samples. As the proof for Theorem B.1, we add and subtract the correct reference quantity $\pi_D$, which can be also denoted as $\pi_D := \pi_D, \pi_D$.*

$$\begin{aligned}
\eta(\pi) - \eta(\pi_{mix}) &= \eta(\pi, \pi) - \eta(\pi_D, \hat{\pi}) \\
&= \eta(\pi, \pi) - \eta(\pi_D, \pi_D) + \eta(\pi_D, \pi_D) - \eta(\pi_D, \hat{\pi}) \\
&= \underbrace{\eta(\pi, \pi) - \eta(\pi_D, \pi_D)}_{L_1} + \underbrace{\eta(\pi_D, \pi_D) - \eta(\pi_D, \hat{\pi})}_{L_3}
\end{aligned}$$

*After $k$ samples, $L_3$ differ in both model and policy, which incorporates both predictor error $\epsilon_m^{N-k}$ and policy error $\epsilon_\pi^{N-k}$ in the last $N - k$ samples. This can be bound by Lemma C.2 with setting $\epsilon_m^{N-k} = \epsilon_m$ and $\epsilon_\pi^{N-k} = \epsilon_\pi$, which results in:*

$$L_3 \geq - \sum_{\tau=k+1}^{N} R_{max}\epsilon_\pi - \sum_{\tau=k+1}^{N} \epsilon_m p(\tau|\widetilde{\theta})$$

*Adding two bounds $L_1$ and $L_3$ together yields the result.*

## C  LEMMAS

**Lemma C.1**  *Policy error:*

$$\max_\tau \left| \prod_{t=1}^{T} \pi_1(a_t|s_t) - \prod_{t=1}^{T} \pi_2(a_t|s_t) \right| \leq \epsilon_\pi \tag{22}$$

*Proof. Considering that $p(s_1)$ and $p(s_{t+1}|s_t, a_t)$ in Eq. (18) are equal to one, we make the following approximation:*

$$\begin{aligned}
D_{TV}(\pi_1(a|s)||\pi_2(a|s)) &= \frac{1}{2} \sum_{s,a} |\pi_1(a|s) - \pi_2(a|s)| \\
&\approx \frac{1}{2} \sum_\tau \left| \prod_{t=1}^{T} \pi_1(a_t|s_t) - \prod_{t=1}^{T} \pi_2(a_t|s_t) \right| \\
&= D_{TV}(p(\tau|\theta_1)||p(\tau|\theta_2))
\end{aligned}$$

*Thus,*

$$\begin{aligned}
\max_s D_{TV}(\pi_1(a|s)||\pi_2(a|s)) &\approx \max_\tau D_{TV}(p(\tau|\theta_1)||p(\tau|\theta_2)) \\
&= \max_\tau \left| \prod_{t=1}^{T} \pi_1(a_t|s_t) - \prod_{t=1}^{T} \pi_2(a_t|s_t) \right| \leq \epsilon_\pi
\end{aligned}$$

**Lemma C.2**  *Expected reward bound:*

$$|\eta(\pi_1) - \eta(\pi_2)| \leq \sum_\tau R_{max}\epsilon_\pi + \sum_\tau \epsilon_m p(\tau|\widetilde{\theta}) \tag{23}$$

*Proof. Here, $\eta(\pi_1)$ denotes the expected true reward of $\pi_1$ with the reward function $R_1$, and $\eta(\pi_2)$ denotes the expected reward of $\pi_2$ with the reward function $R_2$. $\max_{\tau \sim \pi_1} |R_1(\tau) - R_2(\tau)| \leq \delta_m$ and $\max_s D_{TV}(\pi_1(a|s)||\pi_2(a|s)) \leq \delta_\pi$. According to Eq. 3, we have:*

$$|\eta(\pi_1) - \eta(\pi_2)| = | \sum_\tau (R_1(\tau)p(\tau|\theta_1) - R_2(\tau)p(\tau|\theta_2))|$$

*According to Lemma C.1, we have:*

a) $\eta(\pi_1) \geq \eta(\pi_2)$. *Since* $\max_{\tau \sim \pi_1} |R_1(\tau) - R_2(\tau)| \leq \delta_m$, *we can get:* $-\delta_m + R_1(\tau) \leq R_2(\tau) \leq \delta_m + R_1(\tau)$. *Then,*

$$
\begin{aligned}
|\eta(\pi_1) - \eta(\pi_2)| &= |\sum_\tau (R_1(\tau)p(\tau|\theta_1) - R_2(\tau)p(\tau|\theta_2))| \\
&\leq |\sum_\tau (R_1(\tau)p(\tau|\theta_1) - R_2(\tau)_{min}p(\tau|\theta_2))| \\
&= |\sum_\tau (R_1(\tau)p(\tau|\theta_1) - (R_1(\tau) - \delta_m)p(\tau|\theta_2))| \\
&= |\sum_\tau R_1(\tau)p(\tau|\theta_1) - \sum_\tau R_1(\tau)p(\tau|\theta_2) + \sum_\tau \delta_m p(\tau|\theta_2)| \\
&\leq \sum_\tau R_1(\tau)|p(\tau|\theta_1) - p(\tau|\theta_2)| + \sum_\tau \delta_m p(\tau|\theta_2) \\
&= \sum_\tau R_1(\tau)|\prod_{t=1}^T \pi_1(a_t|s_t) - \prod_{t=1}^T \pi_2(a_t|s_t)| + \sum_\tau \delta_m p(\tau|\theta_2) \\
&\leq \sum_\tau R_{max}|\prod_{t=1}^T \pi_1(a_t|s_t) - \prod_{t=1}^T \pi_2(a_t|s_t)| + \sum_\tau \delta_m p(\tau|\theta_2) \\
&\leq \sum_\tau R_{max}\delta_\pi + \sum_\tau \delta_m p(\tau|\theta_2)
\end{aligned}
$$

b) $\eta(\pi_1) < \eta(\pi_2)$. *Since* $\max_{\tau \sim \pi_1} |R_1(\tau) - R_2(\tau)| \leq \delta_m$, *we obtain:* $-\delta_m + R_2(\tau) \leq R_1(\tau) \leq \delta_m + R_2(\tau)$. *Then,*

$$
\begin{aligned}
|\eta(\pi_1) - \eta(\pi_2)| &= |\eta(\pi_2) - \eta(\pi_1)| \\
&= |\sum_\tau (R_2(\tau)p(\tau|\theta_2) - R_1(\tau)p(\tau|\theta_1))| \\
&\leq |\sum_\tau (R_2(\tau)p(\tau|\theta_2) - R_1(\tau)_{min}p(\tau|\theta_1))| \\
&= |\sum_\tau (R_2(\tau)p(\tau|\theta_2) - (R_2(\tau) - \delta_m)p(\tau|\theta_1))| \\
&= |\sum_\tau R_2(\tau)p(\tau|\theta_2) - \sum_\tau R_2(\tau)p(\tau|\theta_1) + \sum_\tau \delta_m p(\tau|\theta_1)| \\
&\leq \sum_\tau R_2(\tau)|p(\tau|\theta_2) - p(\tau|\theta_1)| + \sum_\tau \delta_m p(\tau|\theta_1) \\
&= \sum_\tau R_2(\tau)|p(\tau|\theta_1) - p(\tau|\theta_2)| + \sum_\tau \delta_m p(\tau|\theta_1) \\
&= \sum_\tau R_2(\tau)|\prod_{t=1}^T \pi_1(a_t|s_t) - \prod_{t=1}^T \pi_2(a_t|s_t)| + \sum_\tau \delta_m p(\tau|\theta_1) \\
&\leq \sum_\tau R_{max}|\prod_{t=1}^T \pi_1(a_t|s_t) - \prod_{t=1}^T \pi_2(a_t|s_t)| + \sum_\tau \delta_m p(\tau|\theta_1) \\
&\leq \sum_\tau R_{max}\delta_\pi + \sum_\tau \delta_m p(\tau|\theta_1)
\end{aligned}
$$

*In summary, we have* $|\eta(\pi_1) - \eta(\pi_2)| \leq \sum_\tau R_{max}\delta_\pi + \sum_\tau \delta_m p(\tau|\widetilde{\theta})$, *where* $\widetilde{\theta} = \theta_2$ *if* $\eta(\pi_1) \geq \eta(\pi_2)$; *otherwise,* $\widetilde{\theta} = \theta_1$.

## D EXPERIMENTAL DETAILS AND RESULTS

### D.1 EXPERIMENTAL DETAILS

**NAS Benchmarks.** NAS-Bench-201 (Dong & Yang, 2020) is a benchmark dataset for NAS algorithms built on image classification tasks, including CIFAR-10, CIFAR-100, and ImageNet-16-120 (Chrabaszcz et al., 2017). CIFAR-10 consists of 60,000 (50,000 training images and 10,000 test images) $32 \times 32$ color images in 10 classes and each class contains 6,000 images. CIFAR-100 has 100 classes, and each class contains 600 images (500 training images and 100 testing images). NAS-Bench-201 provides a cell-based search space, where a cell is represented by a directed acyclic graph (DAG). A DAG contains 4 nodes and 6 edges, and each edge has 5 representative operation candidates, which results in 15,625 neural cell architectures in total. Each architecture contains full training logs, validation accuracy, and test accuracy on CIFAR-10, CIFAR-100, and ImageNet-16-120. In summary, NAS-Bench-201 allows researchers to easily compare different approaches by providing all architecture evaluation results.

NAS-Bench-ASR (Mehrotra et al., 2020) is a tabular NAS benchmark for automatic speech recognition. The search space consists of 8242 unique models trained on TIMIT dataset. Each model includes all kinds of runtime measurements, such as the per epoch validation and final test metrics, *Phoneme Error Rate* (PER), and CTC loss. The search space consists of four nodes, with three main edges that can take on one of six operations, and six skip connection edges, which can be set to on or off.

**Baselines.** We compare our method with two types of state-of-the-art methods: traditional NAS algorithms and predictor-based NAS algorithms:

1. Traditional NAS algorithms include: random search (RS) (Bergstra & Bengio, 2012), REA (Real et al., 2019), REINFORCE (Williams, 1992), and BOHB (Falkner et al., 2018). We use the code provided by Dong et al. (2021) to implement these algorithms. To ensure a fair comparison, we also require the search strategy of the baselines sample unique architectures as our method does.

2. Predictor-based NAS algorithms include: MLP (White et al., 2021a), GCN (Wen et al., 2020), BANANAS (White et al., 2021a), BONAS (Shi et al., 2020), NAO (Luo et al., 2018), SemiNAS (Tang et al., 2020), Transformer (Lu et al., 2021) and XGBoost (Chen & Guestrin, 2016). We compare the most representative performance predictors to select the best performance predictor as our predictor. In addition, we combine these performance predictors except Transformer with two widely used search strategies, BO and EA, as the predictor-based NAS algorithms. We use the code from (White et al., 2021b) and NAS-BENCH-SUITE (Mehta et al., 2022) to implement these algorithms .

**Implementation details.** Our method consists of two modules, an RL agent as the search strategy and a performance predictor. The agent consists of input and output layers and a LSTM model. The input-layer is an embedding layer, and the size of each embedding vector is 32. The LSTM model is a two-layer LSTM with 35 hidden units on each layer. The output-layer is a linear layer with 32 hidden units. The agent is trained with the Adam optimizer with the learning rate 0.001. Weights of the agent are initialized uniformly between -0.1 and 0.1. We use a tanh of 2.5 and a temperature of 5.0 for the sampling logits (Bello et al., 2017) to prevent premature convergence and add the controller's sample entropy to the reward, weighted by 0.0001. Especially, the batch size $N$ is 20, which is different from the setting of Zoph & Le (2016). We set $k$ to 2, 5, 2, and 2 for CIFAR-10, CIFAR-100, ImageNet-16-120, and TIMIT, respectively. The initial architecture-accuracy pairs for PNASM and PNASM-A is 100 across all experiments. (Note that following Dong et al. (2021), we train candidate architectures in 12 epochs and retrieve the best architecture by training in full epochs on NAS-Bench-201.)

### D.2 EFFECTIVENESS STUDY ON COMPONENTS OF PNASM.

To figure out which part of PNASM boost the improvement, we conduct a series of ablation experiment on PNASM, which mainly consists of LSTM-based agent, a predictor and the batch strategy. Table 5 shows comparison results of different PNASM variants on NAS-Bench-201 (We also conduct the same experiment on NAS-Bench-ASR, see Table 7). The four variants are:

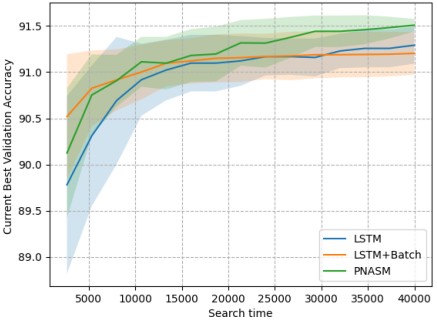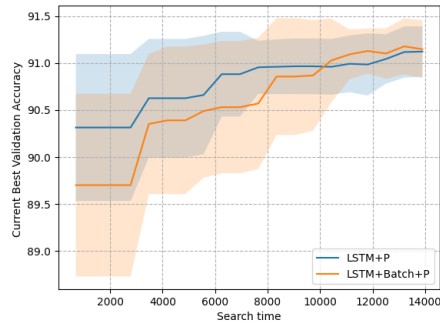

Figure 5: Performance of different variants of PNASM on CIFAR-10. Left figure presents the comparison results among "LSTM","LSTM+Batch" and PNASM. Right figure shows the comparison results between "LSTM+P" and "LSTM+Batch+P", because the search time of the two variants is inconsistent with others.

1. "LSTM" denotes that we run the search just using LSTM-based agent, without a predictor. The agent (policy) is updated after every candidate.

2. "LSTM+Batch" denotes that we update LTSM-based agent (without the predictor) after a batch of candidates is found.

3. "LSTM+P" denotes that we run the search with LSTM-based agent combined with a predictor. The agent is updated after every architecture is found.

4. "LSTM+Batch+P" denotes that we use LSTM-based agent and a predictor to search, but perform batch-wise update of the agent, without evaluating any additional architectures from scratch.

5. PNASM denotes that we use LSTM-based agent and a predictor but perform *mixed batch* update of the agent.

Predictors in all variants are initialized by 100 sampled architectures. We randomly run 20 times for each variant and use the search time as the time budget. Since "LSTM+P" and "LSTM+Batch+P" use the predictor for each candidate, the time cost of evaluating architecture is low. To ensure fair experiments, we allow "LSTM+P" and "LSTM+Batch+P" to sample 2000 architectures, which is greater than that by other variants. From Table 5, we can make the following observation: 1) Batch strategy does not improve LSTM-based agent's performance on CIFAR-10, which is opposite on TIMIT. 2) Although the predictor can reduce the search time, but the long and continuous usage brings large error to the search strategy, as indicated by the results of "LSTM+P" and "LSTM+Batch+P". 3) *Mixed batch* helps LSTM-based agent to make better use of the predictor, since ground-truth data corrects the error caused by long-term usage. Additionally, to better demonstrate the search process of all variants, Figure 5 plots the current best validation accuracy of all variants over the search time on CIFAR-10.

Table 5: Comparisons of different Variants of PNASM on CIFAR-10.

| Method | Search(s) | Validation | Test |
|---|---|---|---|
| LSTM | 40000 | $91.29 \pm 0.19$ | $94.03 \pm 0.24$ |
| LSTM+Batch | 40000 | $91.21 \pm 0.23$ | $94.01 \pm 0.24$ |
| LSTM+P | 15000 | $91.16 \pm 0.24$ | $93.96 \pm 0.24$ |
| LSTM+Batch+P | 15000 | $91.14 \pm 0.30$ | $93.94 \pm 0.27$ |
| PNASM | 40000 | $\mathbf{91.51 \pm 0.05}$ | $\mathbf{94.33 \pm 0.06}$ |

### D.3 ADDITIONAL EXPERIMENTAL ON NAS-BENCH-ASR

#### D.3.1 CHOOSE PREDICTOR FOR NAS-BENCH-ASR

Since the predictor usually has poor generalization ability across the search space, we choose a high-performance predictor for TIMIT on NAS-Bench-ASR again. Table 6 compares the spearman correlation of PER of four predictors: MLP, NAO, SemiNAS, and XGBoost. We randomly run 20 times for each predictor. We can easily make the following observation from Table 6: 1) Predictors show poor generalization ability across the search spaces. For example, SemiNAS performs well on NAS-Bench-201 in Table 1, but fails on NAS-Bench-ASR. 2) MLP performs well if it has sufficient initial training data. 3) XGBoost performs good across the different settings of initialization. According to the experimental results, we choose XGBoost as the predictor on NAS-Bench-ASR.

Table 6: Spearman Correlation of Predictors on TIMIT.

| Training Samples | 100 | 200 | 400 | 600 | 800 |
|---|---|---|---|---|---|
| Test Samples | 200 | 200 | 200 | 200 | 200 |
| MLP | $0.464 \pm 0.08$ | $0.512 \pm 0.05$ | $0.614 \pm 0.06$ | $\mathbf{0.635 \pm 0.07}$ | $\mathbf{0.646 \pm 0.04}$ |
| NAO | $0.417 \pm 0.05$ | $0.446 \pm 0.03$ | $0.475 \pm 0.08$ | $0.482 \pm 0.06$ | $0.477 \pm 0.05$ |
| SemiNAS | $0.418 \pm 0.04$ | $0.435 \pm 0.08$ | $0.466 \pm 0.05$ | $0.502 \pm 0.06$ | $0.495 \pm 0.07$ |
| XGBoost | $\mathbf{0.572 \pm 0.05}$ | $\mathbf{0.590 \pm 0.07}$ | $\mathbf{0.628 \pm 0.07}$ | $0.611 \pm 0.04$ | $0.640 \pm 0.04$ |

#### D.3.2 VARIANTS OF PNASM ON NAS-BENCH-ASR.

Table 7 shows the comparisons of different variants of PNASM on NAS-Bench-ASR. Predictors take 100 true architecture-val pairs to initialize. We can make the similar observation as Table 5 except for the batch strategy. On NAS-Bench-ASR, Batch strategy can help LSTM-based agent to achieve lower validation PER. We speculate that batch strategy is highly related to statistics of dataset on NASBench.

Table 7: Comparisons of different components of PNASM on NAS-Bench-ASR.

| Method | Sampled True Archs | Total Sampled Archs | Val. PER(%) | Test PER(%) |
|---|---|---|---|---|
| LSTM | 300 | 300 | $19.21 \pm 0.13$ | $21.82 \pm 0.29$ |
| LSTM+Batch | 300 | 300 | $19.18 \pm 0.10$ | $21.85 \pm 0.29$ |
| LSTM+P | 100 | 2000 | $19.25 \pm 0.07$ | $21.87 \pm 0.23$ |
| LSTM+Batch+P | 100 | 2000 | $19.21 \pm 0.14$ | $21.91 \pm 0.21$ |
| PNASM | 300 | 1500 | $\mathbf{19.09 \pm 0.10}$ | $\mathbf{21.73 \pm 0.21}$ |

