# OpenReview forum: "How Predictors Affect Search Strategies in Neural Architecture Search?"
_ICLR.cc/2023/Conference — Submitted to ICLR 2023_

### Official Review · Reviewer_9HrM · 2022-10-20

**Confidence:** 4
**Correctness:** 3
**Technical Novelty And Significance:** 2
**Empirical Novelty And Significance:** 2
**Recommendation:** 5

**Clarity, Quality, Novelty And Reproducibility:**

Novelty. The novelty in light of the linked paper is rather limited. The mixed batch construction and the introduced hyperparameter as well as the (short) theoretic analysis are what is really novel about this paper.

Reproducibility. I am convinced that the results are reproducible. I very much appreciate the analysis of the natural edge cases of the hyperparameter.

Clarity. There are quite a few things in the Weakness-Section, that should be addressed to have a more convincing argument and placement of the paper.

Quality. The scientific procedure is sound.

**Strength And Weaknesses:**

### Strengths:

1. The theoretic analysis on the sources of error is nice and basing the method off-of that is a good idea.
1. I presume, that formulating the archtiecture as a policy (albeit I’d strongly encourage to elaborate on that) is a beneficial way to express the architecture, as an incremental buildup on the architecture and attributing reward to model components and their sequential composition is akin to how humans would go about finding new architectures and may be helpful to reduce overal cost. The upside of this is that this heuristic is performance driven and the compositional strategy is being optimized.
1. Having an ablation on the introduced hyperparameter.

### Weaknesses:

1. Considering https://arxiv.org/pdf/1904.02642.pdf, the proposed approach corresponds to an RL based acquisition strategy in a NAS search space, that (just like the linked approach) is allowed to update the surrogate model in its sequential acquisition process. The main difference is that for the policy update, both ground turth architecture evaluations and predictor based evaluations are mixed. The novelty is limited to the construction of the batch and the theoretic analysis. I do not see a clear benefit of having a mixed batch (which is also very costly due to the truth evaluation) over said paper other than speeding up the training of the acquisition function (RL agent) by making use of predicted reward signals as cheap proxy. Just like in the linked paper, the surrogate becomes a moving target by virtue of newly collected trajectories, that takes into account all what can be known in terms of ground truth and is guaranteed to be informative eventually by the sequential process.
1. Fitting a supervised model to the observed rewards as a surrogate model is basically nothing new (as surrogates are doing that). Also the method is rather simplistic in the sense that it merely introduces a hyperparameters for the composition of “true” & “predictor” datapoints which is arbitrarily set albeit having some Ablation on that hyperparameter for their specific problem instances.
1. There are still quite a few language issues (regarding declination etc.) that disturb the flow of reading.
1. At least in the BO literature, one would rather prefer “surrogate” over predictor. In RL one would consider the predictor to be a “model”. This is just a rephrasing but makes it clearer. In the same sense, “predictor-RL-based NAS” should rather be reduced to model-based NAS using an RL controller or something similar. To facilitate readers from other fields the entry to your paper, this might be helpful footnote.

#### Specifically:

1. (Abstract) “Unfortunately, even a promising performance predictor may suffer from the accuracy decline due to long-term and  continuous  usage,  thus  leading  to  the  degraded  performance  of  the  search strategy” seems to be a somewhat wrong or at least insufficient description of what you seek to improve upon. Reading this the first time, it was absolutely unclear to me what exactly you were set out to do (what your issue is). Instead - from my understanding, you could rephrase it like this:
Using the fixed target predictor to an RL controller necessarily will cause performance degradation on new datapoints, as it does not allow for observations updating the predictor and is limited by its precision and mislead by its generalization error. This can be observed in k=0 (4.4). On the other hand k=N is the other extreme case where we only ever collect new datapoints - making it prohibitive to train brittle and expensive RL methods on it.
1. (2.1.) The distinction between training-based & training-free predictors is poorly disambiguated. In particular, any of the methods described here can be used as feature space to a predictor, which then learns how to relate the feature space to a performance measure. After all both categories aim at linking their “statistics” of the architecture - be it the graph or cell representation or the e.g. zero-shot statistics to a performance score.
1. I do not see the difference between a predictor-based NAS (having an encoder & regressor) to a surrogate model such as a GP, that defines a specific kernel (encoder).
1. (3.2) Stating an MBRL problem is not sufficient on its own, but a specification of all its components in your setup-interpreation would bring value in understanding how the model is constructed.
1. (3.2) I would very much appreciate a more detailed outline of how a trajectory corresponds to a description of a neural architecture.
1. (4.4) k=all is informal. Make it k=N, Figure 2. The variation over seeds would be more indicative of the actual performance.
1. At least to me, the depiction of Algorithm 1 is not required.




**Summary Of The Paper:**

The proposed approach corresponds to an RL based acquisition strategy in a NAS search space, that is basing its trajectory on both the observed and predicted (surrogate) evaluations. This is called a mixed batch, whose share in either is a relevant hyperparameter to the success of the model by balancing the prediction error from a surrogate with the computational cost of a full evaluation. This trade-off is identified by a theoretic decomposition of the RL-based acquisition error, comprising of the generalization error of the predictor - to which an unmixed-ground-truth policy may overfit - and the policy error induced by a reward signal of the predictor on the other hand.

**Summary Of The Review:**

Considering the limited novelty and the insufficiently precise formulation of the problem & problem setting I would recommend not accepting the paper in its current form.

---

> ### Author Response · Authors · 2022-11-19
> **Explaination of difference between BO and RL**
>
> We thank the reviewer for the comments concerning our manuscript.  Those comments are all valuable and very helpful for revising and improving our paper, as well as the important guiding significance to our research.
>
> Concern 1.	Considering https://arxiv.org/pdf/1904.02642.pdf, the proposed approach corresponds to an RL based acquisition strategy in a NAS search space, that (just like the linked approach) is allowed to update the surrogate model in its sequential acquisition process.
>
> Response: We don’t agree that our approach is an RL based acquisition strategy in a NAS search space， that is allowed to update the surrogate model in its sequential acquisition process.
> In BO models, the surrogate function is a probabilistic model which captures the updated belief about the objective function f(x). The acquisition function decides which point to sample based on the surrogate model.
> In RL, the agent acts like both the surrogate function and acquisition function in BO, which implicitly models the distribution of f(x) and decides the next point to sample based on the distribution.
> It is noted that for BO-based NAS methods, after sampling a new architecture x, it still requires evaluating its performance f(x) by training from scratch. And the evaluated results are used to update the surrogate function. Surrogate function is just a distribution of f(x), not a good evaluation of f(x). Therefore, both BO and RL-based NAS methods need to train architectures from scratch to evaluate their performance.
> Training architectures from scratch incur a high computational cost. To alleviate that, researchers use a performance predictor to predict the performance.
> In summary, the surrogate model is not a predictor. The surrogate model in BO cannot be used to evaluate the sampled architecture. The sampled architecture is evaluated either by training from scratch or by a predictor in most research.
>
> Concern 2.	 I do not see a clear benefit of having a mixed batch (which is also very costly due to the truth evaluation) over said paper other than speeding up the training of the acquisition function (RL agent) by making use of predicted reward signals as cheap proxy. Just like in the linked paper, the surrogate becomes a moving target by virtue of newly collected trajectories,  that takes into account all what can be known in terms of ground truth and is guaranteed to be informative eventually by the sequential process.
>
> Response: As we indicated in our paper, even a promising performance predictor may suffer from an accuracy decline if we use it during the whole search process without updating it. The inaccurate prediction will affect the search strategy. Mixed batch is a simple procedure to update the predictor based on the analysis of the impact of the predictor on the search strategy.
>
> Concern 3.	I would very much appreciate a more detailed outline of how a trajectory corresponds to a description of a neural architecture.
>
> Response: Please refer to https://arxiv.org/pdf/1611.01578.pdf of how a trajectory corresponds to a description of neural architecture.

---

> > ### Comment · Reviewer_9HrM · 2022-12-05
> > **Thanks for your reply**
> >
> > Hi,
> >
> > Thank you very much for your reply. Overall, I see where you are coming from, but I still don't agree with your view on RL. Therefore, I stay with my overall score.

---

### Official Review · Reviewer_qrWv · 2022-10-25

**Confidence:** 5
**Correctness:** 3
**Technical Novelty And Significance:** 3
**Empirical Novelty And Significance:** 3
**Recommendation:** 5

**Clarity, Quality, Novelty And Reproducibility:**

The paper is good in clarity, novelty and reproducibility. But I am not sure about its quality due to its motivation, which is not supported by sufficient evidence.

**Strength And Weaknesses:**

Strengths:
1.	The paper is well written and easy to understand.
2.	The theoretical and empirical study of predictors on NAS is good.
Weaknesses:
1.	The first question is that the evidence of the motivation is not direct. Since the problem to be solved is that “a predictor suffers from the accuracy decline due to long-term and continuous usage”, the authors need to plot a figure about the decline in accuracy of a predictor over time (search steps) in different settings to support their claim.
2.	Another question is why choose k = 2, 5, 2 in cifar-10, cifar-100, imagenet-16-120 in Table 1, while the result in Table 3 shows that the best k should be 5, 8, 2 ? The best results of the two tables do not seem to match.
3.	Is there any related work about the mixed-batch method?


**Summary Of The Paper:**

The paper proposes a predictor-based neural architecture search with N-sized mixed batch, in which the performances of k architectures are evaluated by training from scratch and the performances of the rest N-k ones are predicted by the predictor. Besides, it studies the impact of predictors on NAS theoretically and empirically.

**Summary Of The Review:**

To summarize, this paper is good in clarity, novelty and reproducibility. However, I am concerned about the weaknesses mentioned above.

---

> ### Author Response · Authors · 2022-11-19
> **New  figure is added as suggested**
>
> We thank the reviewer for the comments concerning our manuscript.  Those comments are all valuable and very helpful for revising and improving our paper, as well as the important guiding significance to our research. We have carefully studied the comments and made corrections which we hope to meet with approval. Revised portions are marked in red in the revised paper.
>
> Concern 1.	The first question is that the evidence of the motivation is not direct. Since the problem to be solved is that “a predictor suffers from the accuracy decline due to long-term and continuous usage”, the authors need to plot a figure about the decline in accuracy of a predictor over time (search steps) in different settings to support their claim.
>
> Response: We have followed the suggestion, and plotted a figure about the decline in accuracy of a predictor over time. Please refer to Figure 1 on page 1 in the revised version.
>
> Concern 2.	Another question is why choose k = 2, 5, 2 in cifar-10, cifar-100, imagenet-16-120 in Table 1, while the result in Table 3 shows that the best k should be 5, 8, 2 ? The best results of the two tables do not seem to match. 3. Is there any related work about the mixed-batch method?
>
> Response: Because the search time is different in the two experiments. Please check the difference of the search time in the two experiments (tables). According to our investigation, we did not find any similar work like mixed-batch in NAS.

---

### Official Review · Reviewer_tJna · 2022-10-25

**Confidence:** 4
**Correctness:** 4
**Technical Novelty And Significance:** 2
**Empirical Novelty And Significance:** 2
**Recommendation:** 5

**Clarity, Quality, Novelty And Reproducibility:**

The paper is well written and is easy to follow.  The contribution is very limited as it applies to surrogate model based RL NAS.

**Strength And Weaknesses:**

Strength:
1. They are able to find well performing architectures in the neural architecture search

Weakness:
1. Is your boost actually owing to your LSTM based RL agent, your batch-wise update policy of the agent or the entire PNASM algorithm? Can you do an ablation study on that? (1) Run the search without a predictor with just the LSTM agent, but update the policy after every candidate. (2) Update the policy after a batch of candidates are found, still without the predictor. (3) Use the agent and the predictor but update after every architecture is found. (4) Use the agent and the predictor but perform batchwise update of the policy but without evaluating any additional architectures from scratch. Rely completely on the predictor performance.

2.  In the PNASM algorithm, the performance predictor is trained on fewer architectures to begin with and is trained as new architectures are encountered. So the contribution of error due to performance predictor is high initially. Including the accuracies of architectures trained from scratch would alleviate it. But if one trained the same performance predictor with more architectures initially, it would lead to a better Kendall Tau score and hence a lower predictor error component. Given that we know the accuracies of the architectures in the NASBench 201 search space, is there a way to analyze these two paths?


**Summary Of The Paper:**

During the search, the neural architecture search (NAS) algorithm uses the validation accuracy of the trained candidate model for feedback. As training a candidate every time is expensive, performance predictors, which take the architecture as input and output the validation accuracy, are used. This paper proposes a trick to improve performance predictor based RL search algorithm.

They use a batched search where the policy samples N architectures, the validation accuracy of all the architectures are obtained to   The RL policy first samples S architectures and the predictor is trained on them. Rather than using this trained predictor to estimate the accuracy of all the sampled architectures at every iteration to perform batch-wise updates of the policy,  the predictor is used to obtain the accuracy of only N-K architectures. The rest of the architectures are trained from scratch. They claim that this alleviates the error contributed in the search owing to the predictor component. They were able to  empirically demonstrate that their technique works.



**Summary Of The Review:**

 We need further justification to bolster the claim that this algorithm reduces the error better than just training the predictor with a lot more data. Also, they need to provide the results of the ablation study that I requested.

---

> ### Author Response · Authors · 2022-11-19
> **Effectiveness Study on Components of PNAS as suggested by reviewer**
>
> We thank the reviewer for the comments concerning our manuscript.  Those comments are all valuable and very helpful for revising and improving our paper, as well as the important guiding significance to our research. We have carefully studied the comments and made corrections which we hope to meet with approval. Revised portions are marked in red in the revised paper.
>
> Concern 1.	Is your boost actually owing to your LSTM based RL agent, your batch-wise update policy of the agent or the entire PNASM algorithm? Can you do an ablation study on that? (1) Run the search without a predictor with just the LSTM agent, but update the policy after every candidate. (2) Update the policy after a batch of candidates are found, still without the predictor. (3) Use the agent and the predictor but update after every architecture is found. (4) Use the agent and the predictor but perform batchwise update of the policy but without evaluating any additional architectures from scratch. Rely completely on the predictor performance.
>
> Response: We have followed the suggestion, and conducted experiments on the four variants. Please refer to Suppl. D.2 Effectiveness Study on Components of PNAS. In Table 5,
> 1) “LSTM” denotes that we run the search just using an LSTM-based agent, without a predictor. The agent (policy) is updated after every candidate;
>  2) “LSTM+Batch” represents that we update the LTSM-based agent (without the predictor) after a batch of candidates is found;
> 3) “LSTM+P” indicates that we run the search with an LSTM-based agent combined with a predictor. The agent is updated after every architecture is found;
> 4) “LSTM+Batch+P” denotes that we use an LSTM-based agent and a predictor to search, but perform a batch-wise update of the agent, without evaluating any additional architectures from scratch. According to the experimental results, the improvement of PNASM is indeed attributed to the “mixed batch”.
>
> Concern 2.	In the PNASM algorithm, the performance predictor is trained on fewer architectures to begin with and is trained as new architectures are encountered. So the contribution of error due to performance predictor is high initially. Including the accuracies of architectures trained from scratch would alleviate it. But if one trained the same performance predictor with more architectures initially, it would lead to a better Kendall Tau score and hence a lower predictor error component. Given that we know the accuracies of the architectures in the NASBench 201 search space, is there a way to analyze these two paths?
>
> Response: You are correct that if we train the same performance predictor with more architectures initially, it would lead to a better Kendall Tau score, as demonstrated in Table 1. But if we initialize the predictor with more samples, it will incur more computational costs. Besides, we cannot guarantee that the prediction is accurate during all search procedure. Thus, considering the trade-off between the performance and the time cost, we suggest initializing the predictor with a proper setting of samples. Our models PNASM and PNASM-A both use 100 true architecture-val pairs to initialize the predictor in all experiments.

---

### Official Review · Reviewer_k5Nf · 2022-10-26

**Confidence:** 4
**Correctness:** 4
**Technical Novelty And Significance:** 2
**Empirical Novelty And Significance:** 2
**Recommendation:** 5

**Clarity, Quality, Novelty And Reproducibility:**

- The paper is in general clearly written and it covers most of the necessary background in and relevant literature in NAS and MBRL.

- The proposed algorithm is relatively simple and the novelty aspect is marginal.

- The authors provide the codebase to reproduce the results

**Strength And Weaknesses:**

The motivation of the paper to investigate the usage of model-based performance predictors for NAS inside a MBRL framework seems plausible. The paper is in general easy to follow and well-structured. There are some interesting theoretical guarantees adopted from the MBRL literature, which the authors built their algorithm, which seems simple and effective. Nevertheless, I have the following criticism:

- **Not enough empirical evaluations**. While it is useful to evaluate a new proposed NAS method on tabular benchmarks such as NAS-Bench-201 used by the authors, I also find it necessary to evaluate on other tabular benchmarks, such as the ones in NAS-Bench-Suite [1], and on a real NAS benchmark.

- **Novelty and limitations**. The proposed algorithm is based on a simple modification of model-based RL, derived from theoretical guarantees adopted from the RL literature. The theoretical justification is interesting but the novelty in the method itself is slightly incremental. Moreover, RL nowadays is not the off-the-shelf choice for NAS, as there are much more efficient methods existing.

-- Questions --

- What is the motivation behind using the specific proposed adaptive method in Section 4.4?

- Would the same Mixed batch method be used for other black-box methods such as evolutionary strategies, or does it work only for MBRL because of the theoretical guarantees?

-- Minor --

Line 3 in Algorithm 1 seems unnecessary.

-- References --

[1] NAS-Bench-Suite: NAS Evaluation is (Now) Surprisingly Easy. Mehta et al. ICLR 2022

**Summary Of The Paper:**

This paper evaluates model based performance predictors in NAS inside a model-based reinforcement learning (MBRL) framework. The authors show theoretically that as long as the expected predicted reward by the performance predictor improves by a certain factor during the optimization, one can guarantee improvements under the expected true reward. The efficacy of the method is demonstrated on the NAS-Bench-201 benchmark, where it outperforms most of the other black-box and predictor based NAS algorithms.

**Summary Of The Review:**

Despite the theoretical guarantees and the motivation, the paper is lacking more empirical evidence  that the method works on a wide range of benchmarks, therefore it needs more work before it is ready for acceptance.

---

> ### Author Response · Authors · 2022-11-19
> **Additional empirical evaluations on NAS-Bench-Suite**
>
> We thank the reviewer for the comments concerning our manuscript. Those comments are all valuable and very helpful for revising and improving our paper, as well as the important guiding significance to our research. We have carefully studied the comments and made corrections which we hope to meet with approval. Revised portions are marked in red in the revised paper.
>
> Concern 1.	While it is useful to evaluate a new proposed NAS method on tabular benchmarks such as NAS-Bench-201 used by the authors, I also find it necessary to evaluate on other tabular benchmarks, such as the ones in NAS-Bench-Suite [1], and on a real NAS benchmark.
>
> Response: We have followed the suggestion, and conducted experiments on NAS-Bench-ASR using the code provided by NAS-Bench-Suite. The experiments include choosing a predictor for NAS-Bench-ASR in Table 6, evaluating the performance of our model and other popular algorithms in Table 3, and ablation study in Table 7. As demonstrated in the experimental results, our model still performs well.
>
> Concern 2.	The proposed algorithm is based on a simple modification of model-based RL, derived from theoretical guarantees adopted from the RL literature. The theoretical justification is interesting but the novelty in the method itself is slightly incremental. Moreover, RL nowadays is not the off-the-shelf choice for NAS, as there are much more efficient methods existing.
>
> Response： We agree that RL nowadays is not the off-the-shelf choice for NAS. But we believe that it can achieve comparable performance to evolutionary algorithms with less computational cost if it is equipped with appropriate usage of the predictor.
>
> Concern 3.	What is the motivation behind using the specific proposed adaptive method in Section 4.4?
>
> Response: The setting of the ratio of true samples to predicted ones in a batch requires precisely fine-tuning. To simplify the setting, we propose an adaptive method to dynamically adjust the ratio by measuring the differences between policies in two consecutive iterations. That is, if the impact of the predictor model is large on policies, which indicates that the large prediction error leads to a large variation in policies, we then increase the ratio of true samples in a batch; otherwise, if the impact of the predictor model is small on policies, which indicates that the prediction error is small, we then can decrease the ration of true samples in a batch to save time.
>
> Concern 4.	Would the same Mixed batch method be used for other black-box methods such as evolutionary strategies, or does it work only for MBRL because of the theoretical guarantees?
>
> Response: This is a very insightful point. Although mixed batch is proposed based on the analysis in the framework of MBRL, we think it is worth trying this method on evolutionary algorithms. We will conduct this experiment in future work.

---

### Decision · Program_Chairs · 2023-01-20

**Decision:**

Reject

**Justification For Why Not Higher Score:**

Most of the reviewers think the paper has limited novelty.

**Justification For Why Not Lower Score:**

N/A

**Metareview: Summary, Strengths And Weaknesses:**

The paper views performance predictor-based NAS as a model-based RL algorithm and uses this connection to establish theoretical analysis as well as propose a mixed batch approach. However, most of the reviewers think the paper has limited novelty. The improvements are also marginal, given that the main results are only obtained under one single setting.